# Interactions of Segmented Filamentous Bacteria (*Candidatus Savagella*) and bacterial drivers in colitis-associated colorectal cancer development

**Annie E. Wolfe**[1]ᵒ, **Jacob E. Moskowitz**[2]ᵒ, **Craig L. Franklin**[1,2,3,4,5]ᵒ, **Timothy L. Wiemken**[6], **Aaron C. Ericsson**[1,2,3,4,5]ᵒ*

1 Molecular Pathogenesis and Therapeutics, University of Missouri, Columbia, Missouri, United States of America, 2 Department of Veterinary Pathobiology, University of Missouri, Columbia, Missouri, United States of America, 3 University of Missouri Metagenomics Center, Columbia, Missouri, United States of America, 4 University of Missouri Mutant Mouse Resource and Research Center, Columbia, Missouri, United States of America, 5 University of Missouri College of Veterinary Medicine, Columbia, Missouri, United States of America, 6 Saint Louis University Center for Health Outcomes Research, St. Louis, Missouri, United States of America

ᵒ These authors contributed equally to this work.
* ericssona@missouri.edu

**Data Availability Statement:** Data may be found in the SRA database under bioproject ID PRJNA548523. The reviewer link is as follows:

## Abstract

Colorectal cancer (CRC) risk is influenced by host genetics, sex, and the gut microbiota. Using a genetically susceptible mouse model of CRC induced via inoculation with pathobiont *Helicobacter* spp. and demonstrating variable tumor incidence, we tested the ability of the Th17-enhancing commensal *Candidatus Savagella*, more commonly denoted as Segmented Filamentous Bacteria (SFB), to influence the incidence and severity of colitis-associated CRC in male and female mice. To document the composition of the gut microbiota during CRC development and identify taxa associated with disease, fecal samples were collected before and throughout disease development and characterized via 16S rRNA sequencing. While there were no significant SFB-dependent effects on disease incidence or severity, SFB was found to exert a sex-dependent protective effect in male mice. Furthermore, SFB stabilized the GM against *Helicobacter*-induced changes post-inoculation, resulting in a shift in disease association from *Helicobacter* spp. to *Escherichia coli*. These data support sex-dependent SFB-mediated effects on CRC risk, and highlight the complex community dynamics within the GM during exposure to inflammatory pathobionts.

## Introduction

Colorectal cancer (CRC) is the second leading cause of cancer-related mortality in the United States. In 2017 alone, there were an estimated 50,260 deaths [1]. Colitis-associated CRC (CAC) is a devastating sequela of inflammatory bowel diseases (IBD) such as Ulcerative Colitis (UC) and Crohn's Disease (CD), recurrent, chronic inflammatory diseases affecting the colon or any

https://dataview.ncbi.nlm.nih.gov/object/
PRJNA548523?reviewer=
9f68h3l705qbmaurmmg3r0b8hu.

**Funding:** C.F. received funding (Grant#: U42 OD010918) from National Institutes of Health (NIH), Office of the Director (https://www.nih.gov/institutes-nih/nih-office-director). A.E. received funding (grant#: K01 OD019924) from NIH, Office of the Director (https://www.nih.gov/institutes-nih/nih-office-director). The funders had no role in study design, data collection and analysis, decision to publish, or preparation of the manuscript.

**Competing interests:** The authors have declared that no competing interests exist.

region of the gastrointestinal tract (GIT) respectively [2]. The risk of developing CAC increases with the duration of IBD, reaching twenty percent after thirty years duration [3]. Although CAC represents less than two percent of cases of CRC, the challenges posed by difficulty of detection and treatment of this subset negatively affect prognosis [4–6].

Inflammation, among other risk factors of CRC, such as diet, smoking, and obesity, have a complex relationship with the gut microbiota (GM) [1–3]. The gut microbiota is the ecosystem of microorganisms inhabiting the GIT, with a profound influence on immune development and function, nutrient absorption and metabolite generation, and susceptibility to diseases of the gut and peripheral systems [7]. IBD-specific models such as the IL-10 knockout mouse have implicated different naturally occurring GM profiles in modulating severity of disease [8]. Genetically identical IL-10 knockout mice on a C57BL/6J background harboring a GM inherited from Taconic mice exhibit more severe inflammation than those harboring a GM originating from Charles River mice [8]. These data demonstrate that different GM profiles among genetically susceptible individuals can significantly affect the severity of intestinal inflammation which contributes to increased risk of CAC. Delineating the mechanisms behind this relationship could help to identify potential therapies to decrease the risk of CAC development in IBD patients.

Mutations affecting the TGF-β pathway, an anti-inflammatory and pro-apoptotic cytokine, are commonly cited as an inciting incident for CRCs [9]. For this reason, Smad3$^{-/-}$ mice, harboring a deletion for the Smad3 signaling molecule downstream of the TGF-β receptor, are often used in combination with pathobiont bacterial species such as *Helicobacter bilis and H. hepaticus* to study colitis-associated CRC [2, 10]. These *Helicobacter* spp. are suspected to act as provocateurs, instigating a host immune response against other commensal bacteria [11, 12], and thus creating an environment of chronic intestinal inflammation as a driver of CRC development. Smad3$^{-/-}$ mice which do not receive this trigger do not develop CRC [10].

However, only 20–66% of *Helicobacter* spp.-inoculated Smad3$^{-/-}$ mice develop CRC by 14 weeks post-inoculation [2, 10]. Because the GM is known to vary from institution to institution [13], and has already been shown to affect disease phenotype of similar models such as the IL-10$^{-/-}$ IBD model [8] and the Pirc rat CRC model [14], we hypothesized that initial static features or subsequent dynamic shifts of the GM following *Helicobacter* spp.-inoculation would modulate disease incidence and severity in the Smad3$^{-/-}$ CRC model.

We also introduced segmented filamentous bacteria (SFB, Candidatus *Savagella*) into our Smad3$^{-/-}$ model due to its status as a keystone species in the modulation of IgA production and Th17 differentiation [15–17], two important components of mucosal inflammatory host responses to the GM. Historically, SFB colonizes most inbred specific pathogen-free (SPF) mice from Charles River Laboratories, Envigo, and many strains from Taconic, but rarely mice from the Jackson Laboratory [16, 17], contributing to variation in disease phenotype in models of IBD [18], Rheumatoid Arthritis [19], type 1 diabetes [20], and multiple sclerosis [21]. Moreover, increased colonization with SFB has been anecdotally associated with ulcerative colitis in people [22], and SFB is believed to enhance induction of Th17 in people as it does in rodents [23].

Thus, using the Smad3$^{-/-}$ model, we sought to identify individual bacteria, or combinations of bacteria, which may contribute to the observed differential susceptibility to CAC. Building on previous findings, we were particularly interested in the composition of the GM prior to, and shortly after, inoculation with *Helicobacter* spp., at the same time that host inflammatory markers are predictive of CAC later in life [2]. Our ultimate objective is a better understanding of the role that SFB and the background GM play in colitis-associated cancer development, in order to develop therapeutic strategies that address and reduce risk of CRC development in IBD patients.

## Results

### Experimental design

An in-house colony of Smad3$^{-/-}$ mice (a gift from Lillian Maggio-Price and originally from the Jackson Laboratory) was divided into two groups of breeding trios and pairs. One group received an inoculation of pure SFB, obtained from endemically colonized BALB/cAnNHsd as previously described [24], while the other group retained its original GM. Breeders were allowed to give birth, thus transferring their gut microbiota vertically to several generations of pups which were used as cohorts in this study. With the understanding that SFB colonization often wanes to undetectable levels over time [25–27], and that the previously reported CRC-predictive spike in pro-inflammatory host mRNAs such as IL-1β occurs as early as one week post-*Helicobacter* spp.-inoculation [2], presence of SFB was confirmed through fecal PCR screening at critical early timepoints from four to five weeks of age following *Helicobacter* spp.-inoculation to assess early events in the development of CAC. Those which tested positive for SFB are referred to as SFB+ and those which did not are SFB-. Pups from SFB+ breeders which tested negative for SFB by PCR were removed from the study.

   SFB+ and SFB- weanling Smad3$^{-/-}$ mice were inoculated by gastric gavage with approximately $1 \times 10^8$ CFU mixture of *Helicobacter hepaticus* and *H. bilis* twice, twenty-four hours apart. Freshly evacuated fecal pellets were collected between 6 a.m. and 7 a.m. at the following timepoints: pre-inoculation (pre), before the second inoculation (mid), Day 1 (D1), D4, D7, 2 weeks (2W), 3W, 5W, 8W, and 14W post-inoculation (PI) (Fig 1A). Cecal contents were also collected at necropsy at 14 weeks PI. Following sacrifice, colonic tissue was analyzed histologically and then scored based on epithelial changes, inflammation, and tumor size and invasiveness (Fig 1B and 1C). DNA from each fecal sample was analyzed via 16S rRNA amplicon sequencing for a snapshot of the GM at each time-point for each mouse, and then retroactively categorized based on lesion scores or the presence/absence of CRC at sacrifice.

   A spike in pro-inflammatory host mRNAs such as IL-1β as early as D7 PI is predictive of CRC development in Smad3$^{-/-}$ mice [2]. For this reason, and in order to assess key changes in the GM before tumor development can contribute to changes in the GM, we focused primarily on the early time-points in this study.

### No SFB-dependent difference in penetrance or severity of CAC

Overall, regardless of SFB status, *Helicobacter* spp.-inoculated mice demonstrated similar disease penetrance (S1A Fig) with 32.8% (19/58) and 24.1% (7/29) of SFB- and SFB+ mice developing histologically identifiable CRC, respectively ($p = 0.562$, Chi square). Similarly, when comparing the tumor score alone (S1B Fig; $p = 0.43$, Kruskal-Wallis ANOVA on ranks), or the overall disease score which also takes into account inflammation and pre-neoplastic epithelial changes (S1C Fig; $p = 0.67$), no differences were detected between SFB- and SFB+ mice.

### No early GM profile found to be predictive of CRC development

To determine whether the GM differed between mice which did or did not eventually develop CAC, data were stratified by time-point and comparisons were visualized and tested via principal coordinate analysis and permutational multivariate analysis (PERMANOVA), respectively. PCoA graphs, based on the weighted Bray-Curtis and Jaccard similarity, of samples collected pre-*Helicobacter* spp.-inoculation, demonstrated no discernible clustering of CRC+ or CRC- mice in either SFB- (S2A Fig) or SFB+ (S2B Fig) groups. Therefore, no predisposing GM profile was detected for mice which do and do not eventually develop CRC in these two colonies.

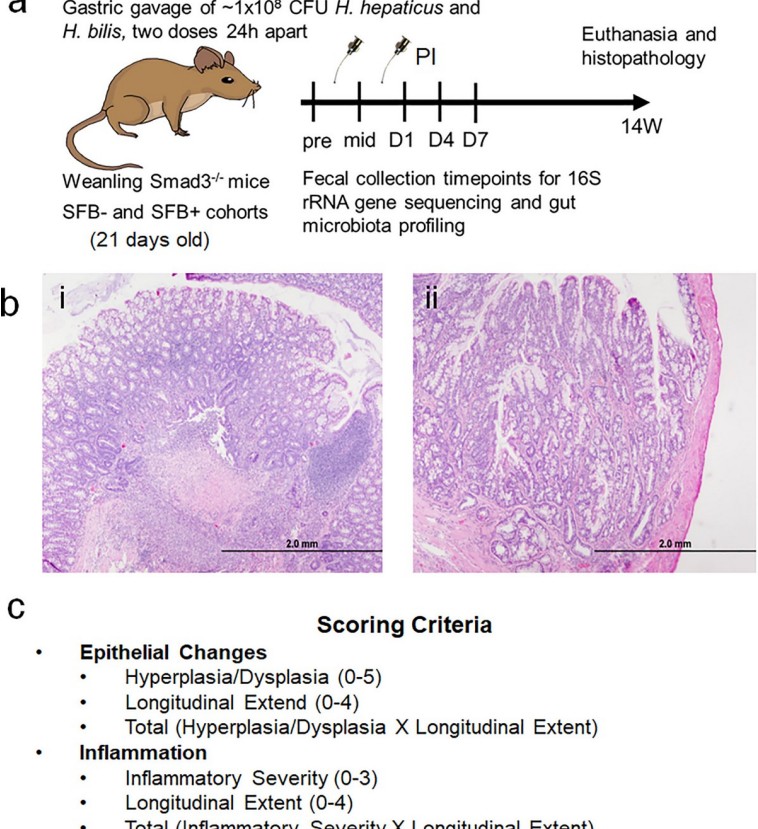

**Scoring Criteria**

- **Epithelial Changes**
  - Hyperplasia/Dysplasia (0-5)
  - Longitudinal Extend (0-4)
  - Total (Hyperplasia/Dysplasia X Longitudinal Extent)
- **Inflammation**
  - Inflammatory Severity (0-3)
  - Longitudinal Extent (0-4)
  - Total (Inflammatory Severity X Longitudinal Extent)
- **Tumor**
  - Tumor Size (#10x fields/tumor)
  - Invasiveness (1-3)
  - Total (Tumor Size X Invasiveness)
- **Overall Lesion Score** (Epithelial Changes + Inflammation + Tumor Totals)

**Fig 1. Experimental design.** A) Two subcolonies of Smad3[-/-] mice were maintained concurrently. SFB+ mice were generated by gastric gavage of Smad3[-/-] breeders with pure SFB inoculum or sham, and subsequent breeding to generate experimental mice. At weaning (21 days old), mice from the SFB- and SFB+ colonies were inoculated via gastric gavage with two doses of *Helicobacter hepaticus* and *Helicobacter bilis* at approximately $1 \times 10^8$ CFUs per inoculum. Fecal samples were collected prior to inoculation (Pre), between the two inoculations (mid), then Day 1 (D1), D4, D7, 2W (two weeks), 3W, 5W, 8W, and 14W post-inoculation (PI). Mice were euthanized at 14 weeks, cecal contents collected, and colon collected for histopathological examination and lesion scoring. B) Representative Haemotoxylin and Eosin (H&E) stained colons of Smad3[-/-] mice sacrificed 14W post-inoculation showing i) severe inflammation of tissue and ii) adenocarcinoma invading into the lamina muscularis, exhibiting branched structure and cell hyperplasia. C) The lesion scoring system for colons in this study, taking into account epithelial changes, inflammation, and tumor scores to generate an overall score. Drawings are property of Annie E. Wolfe, an author of this paper.

Similar analyses were performed using data from all early time-points post-inoculation with the same results.

## Interactions between sex and SFB influence incidence and severity of CRC

When separated on the basis of sex within the SFB- and SFB+ groups, intriguing interactions were revealed. Firstly, CRC incidence in SFB+ male mice was significantly (p = 0.01, Fisher's Exact test) reduced compared to SFB+ female mice (Fig 2A). A similar trend, albeit not statistically significant, was seen in SFB- mice, suggesting a possible sex bias in this model. Similarly,

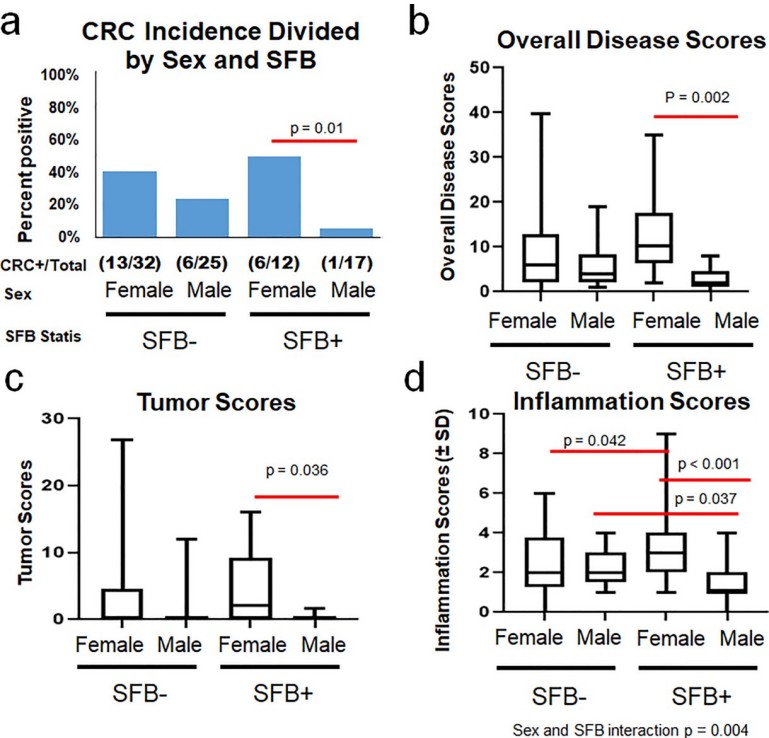

**Fig 2. Sex and SFB interactions influence incidence and severity of CRC.** A) CRC incidence of SFB- and SFB+ male and female *Helicobacter* spp.-inoculated Smad3$^{-/-}$ mice. Fisher's Exact Tests within SFB- and SFB+ category (to narrow down an overall chi square test *p* = 0.0226) revealed significant differences in incidence of CRC between males and females in SFB+ group (*p* = 0.01) but not within the SFB- group (*p* = 0.16). B) Overall Disease Scores C) Tumor Scores, and D) Inflammation Scores of male and female mice within the SFB- and SFB+ categories, analyzed via Two-Way ANOVA (*post hoc* Holm-Sidak). Significance and corresponding p values indicated by red lines between different groups. N values for each group indicated in panel A.

overall disease scores (taking into account epithelial changes, inflammation, and tumor size and invasiveness), reflect this pattern of reduced disease severity in SFB+ male mice compared to female counterparts (Fig 2B). Tumor scores follow a similar pattern (Fig 2C). Of particular note, inflammation scores, taken alone, exhibit a significant interaction between sex and SFB (p = 0.004, Two-Way ANOVA), wherein SFB+ female mice have more severe inflammation than SFB- females but SFB+ males have less severe inflammation than SFB- males (Fig 2D). The cause of this polarization of disease severity between male and female mice in response to SFB colonization is as yet unknown, but an area of interest for continued research.

Following concerns of a cage effect, a linear mixed modeling statistical analysis was performed on tumor and overall disease severity, with emphasis on cage clustering within the SFB- and SFB+ groups. These analyses found an overall decrease in male tumor (p = 0.0048) and overall disease (p = 0.00089) scores compared to female scores with an intracluster correlation coefficient (ICC) score of 0.076 and 0.074 respectively. The ICC, which spans from 0 to 1, measures the likelihood of a cage or group effect skewing the data. Scores closer to 1 call for the need to analyze each cage or grouping as one individual in statistical analysis.

PCoAs using Bray-Curtis indices (S3 Fig) examine combinations of SFB status, sex, and CRC development on the pre-inoculation timepoint. Taking into account cage groupings in SFB- and SFB+ pre-inoculation PCoAs of mice which did and did not develop CRC, CRC + mice are well distributed throughout the groups and do not cluster by CRC status or within cage assignments. The GM profiles of SFB+ mice and SFB- mice do show significant

differences overall as groups. However, within the SFB+ category, male and female mice GM profiles are not significantly different.

## SFB impacts the GM of Smad3$^{-/-}$ mice during disease development

Stacked bar charts portraying the relative abundance of operational taxonomic units (OTUs) in each group (averaged) provide a subjective overview of the GM over time during disease progression (Fig 3A). Three patterns emerged from pre-inoculation to 2W PI within the SFB-CRC-, SFB-CRC+, SFB+CRC-, and SFB+CRC+ groups. First, the GM of SFB+ mice, regardless of CRC development, remained relatively consistent from pre to 2W post-inoculation when compared to the GM of SFB- mice. Second, colonization with *Helicobacter spp.* (orange-yellow) peaked at D4 in SFB- mice regardless of CRC development, but was blunted in SFB+ mice. Lastly, microbes in the family *Enterobacteriaceae* (more specifically resolved as *Escherichia-Shigella Escherichia* sp.) (light blue) bloomed concurrently with *Helicobacter* spp.

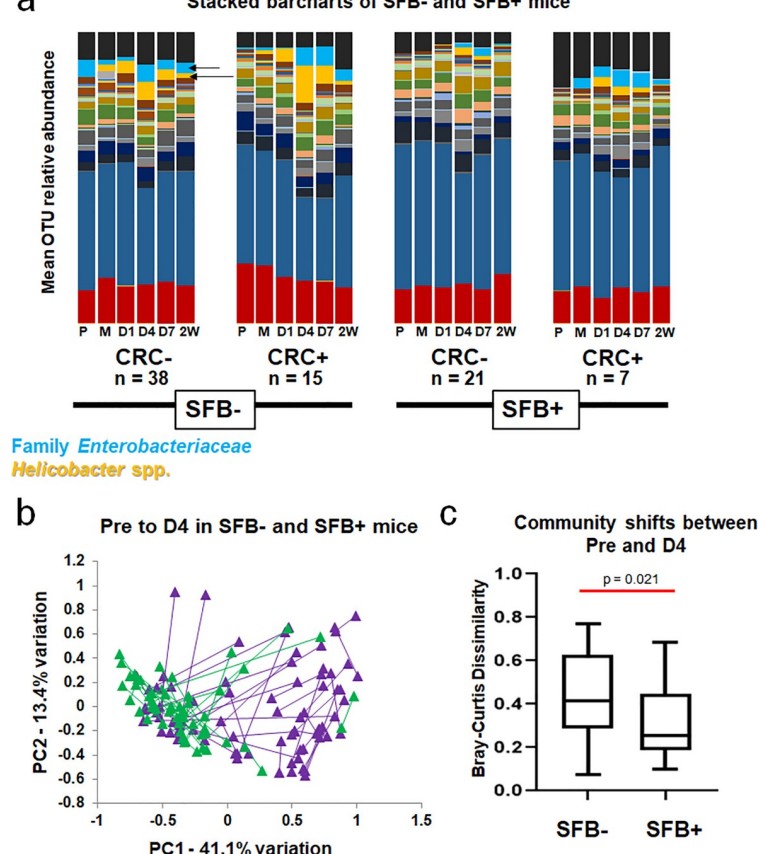

**Fig 3. SFB stabilizes the GM and prevents large shifts in GM structure following *Helicobacter* spp. inoculation.** A) Relative abundance of OTUs within CRC+ and CRC- mice in SFB- and SFB+ groups, shown as group means at pre-inoculation (P), after the first inoculation (M), and at day 1 (D1), D4, D7, and two weeks (2W) PI. Yellow-orange represents *Helicobacter* spp. (including *Helicobacter ambiguous taxa* and *Helicobacter uncultured bacterium*) and light blue represents Family *Enterobacteriaceae* (more specifically denoted as *Escherichia-Shigella Escherichia* sp.). These are indicated by arrows. B) Principal Component Analysis (PCA) of Pre and D4 samples of each mouse in SFB+ (green) and SFB- (purple) groups. Length of line denotes overall change in composition from Pre to D4 in each mouse. C) Mean (±SD) shifts in GM community shown in B, using Bray-Curtis Dissimilarity Index. Mann-Whitney Rank Sum test *p* = 0.021.

in all groups but SFB+CRC-. S4 Fig contains the full legend for the barcharts provided in Fig 3A.

## SFB stabilizes the gut microbiota following *Helicobacter spp.*-inoculation

Subjectively, SFB- mice appear to have greater community shifts following *Helicobacter* spp.-inoculation than SFB+ mice. Fig 3B shows a weighted PCoA of Pre and D4 samples from all mice, with lines connecting samples from the same mouse. These distances, representing the dissimilarity between Pre and D4 GM composition of each pair of samples, were used to generate the intra-subject Bray-Curtis dissimilarity index for each mouse using PAST software [28]. Each subject's Bray-Curtis Dissimilarity index value were then averaged by group and analyzed (Fig 3C). On average, SFB- Pre and D4 time-points were more dissimilar (higher Bray-Curtis dissimilarity value) than Pre and D4 time-points of SFB+ mice ($p = 0.023$, Mann-Whitney rank sum test), indicating a greater shift in community structure following *Helicobacter* spp.-inoculation in SFB- mice. In conclusion, despite similar overall penetrance and severity of CRC between SFB- and SFB+ mice, the presence of SFB is associated with a stabilizing effect on the gut microbiota, preventing the dysbiosis often accompanying and contributing to CRC development. No differences in GM stability on the basis of sex were seen in either SFB + or SFB- group, though there was a trend ($p = 0.066$, Two-Way ANOVA, *post hoc* Holm-Sidak) toward greater stability in male mice relative to female mice, regardless of SFB status. No interactions were detected between SFB status and Sex (S5 Fig).

## Presence of SFB alters which taxa correlate with disease severity

Spearman's rank correlations taking into account overall disease score and relative abundance of family (Table 1) and OTU (Table 2) relative abundance in SFB+ and SFB- mice at Pre, D4, and D7 PI were performed to better understand which specific taxa are associated with disease severity. Negative R values are negatively correlated with disease severity, and positive R values are positively correlated with disease severity. Bolded taxa contain an r value stronger than or equal to ±0.5. As with anything involving multiple testing at a high scale, these should be taken with a grain of salt, especially with weak overall r values (averaging around ±3 and ±4). While members such as family *Akkermansiaceae* and *Helicobacteriaceae* correlate with disease severity in SFB+ mice only, families associated with mitochondria and Bromus Tectorum (a plant-based sequence likely from feed), which are likely meaningless to CRC development, also appear. However, the complete lack of overlap between correlative taxa between SFB+ and SFB- mice is intriguing, and suggestive that different bacterial taxa are playing a role or responding to disease severity depending on the presence of SFB. In the case of family *Akkermansiaceae* and, more specifically, *Akkermansia uncultured bacterium*, relative abundance is positively correlated with disease severity in SFB- mice but negatively correlated with disease severity in SFB+ mice. Notably, in SFB+ mice, family *Prevotellaceae*, specifically *Prevotella 9 uncultured bacterium*, and family *Desulfovibrionaceae* are strongly negatively correlated with disease severity. Family *Helicobacteriaceae*, which can include other species than the inoculated *H. hepaticus* and *H. bilis*, is positively correlated with disease in SFB+ mice at D7. However, at the taxa level, *Helicobacter unc. bacterium* is positively correlated with SFB- mice while *Helicobacter ambiguous taxa* is positively correlated with SFB+ mice.

A hypothesis-driven approach would be needed to better understand how these different bacteria are either driving or responding to disease within SFB- and SFB+ contexts, but these data suggest that taxa important to disease development in this model differ based on the presence or absence of SFB.

**Table 1. Spearman's ranked correlation of bacterial families at Pre, D4, and D7 in SFB+ and SFB- mice.**

| Timepoint | GM | Family | R value | P value |
|---|---|---|---|---|
| **Pre** | **SFB+** | **Family Desulfovibrionaceae** | **-0.643** | **0.000218** |
| **Pre** | **SFB+** | **Family Prevotellaceae** | **-0.527** | **0.00413** |
| Pre | SFB+ | Family Ruminococcaceae | -0.45 | 0.0167 |
| Pre | SFB+ | Family Burkholderiaceae | 0.467 | 0.0124 |
| **D4** | **SFB+** | **Family Prevotellaceae** | **-0.682** | **0.0000453** |
| **D4** | **SFB+** | **Family Rikenellaceae** | **-0.561** | **0.00203** |
| D4 | SFB+ | Family Lachnospiraceae | -0.458 | 0.0145 |
| D4 | SFB+ | Family Burkholderiaceae | 0.4 | 0.0349 |
| **D4** | **SFB+** | **Family Helicobacteriaceae** | **0.509** | **0.00588** |
| **D4** | **SFB+** | **Family Erysipelotrichaceae** | **0.6** | **0.00078** |
| **D7** | **SFB+** | **Family Prevotellaceae** | **-0.666** | **0.0000689** |
| **D7** | **SFB+** | **Family Desulfovibrionaceae** | **-0.504** | **0.00558** |
| **D7** | **SFB+** | **Family Ruminococcaceae** | **-0.498** | **0.00619** |
| D7 | SFB+ | Family Lachnospiraceae | -0.491 | 0.00712 |
| D7 | SFB+ | Mitochondria | 0.369 | 0.0487 |
| D7 | SFB+ | Family Erysipelotrichaceae | 0.379 | 0.0424 |
| **D7** | **SFB+** | **Family Burkholderiaceae** | **0.501** | **0.0059** |
| **D7** | **SFB+** | **Bromus Tectorum** | **0.536** | **0.00288** |
| **D7** | **SFB+** | **Family Akkermansiaceae** | **0.55** | **0.00212** |
| **D7** | **SFB+** | **Family Helicobacteriaceae** | **0.551** | **0.00208** |
| Pre | SFB- | Family Burkholderiaceae | -0.452 | 0.00405 |
| Pre | SFB- | Clostridiales vadinBB60 | -0.403 | 0.0112 |
| Pre | SFB- | Family Streptococcaceae | 0.324 | 0.0442 |
| Pre | SFB- | Family Bacteroidaceae | 0.373 | 0.0197 |
| D4 | SFB- | Family Akkermansiaceae | -0.375 | 0.00971 |
| D7 | SFB- | Family Akkermansiaceae | -0.34 | 0.017 |
| D7 | SFB- | Family Fusobacteriaceae | 0.333 | 0.0197 |

Bacterial families which are significantly correlated with high or low overall disease scores are listed for Pre, D4, and D7 in both SFB- and SFB+ mice. R values and P values are listed side by side. R values range from -1 to +1, with strong negative and positive correlations (respectively), closer to ±1 than 0. Bolded values indicate stronger correlations.

### Relative abundance of *Helicobacter* spp. at D4 PI predictive of CRC development in SFB- but not SFB+ mice

In addition to the overall temporal uniformity of the GM in the presence or absence of SFB, we were also interested in the influence of SFB on *Helicobacter* spp. Notably, the proliferation of *Helicobacter* spp. by D4 differed between SFB- and SFB+ mice (Fig 4A). The relative abundance of *Helicobacter* spp. in SFB+ mice followed similar kinetics regardless of CRC development (Fig 4A). However, at D4 PI, SFB- mice harbored significantly greater (p = 0.016, Two-Way RM ANOVA) relative abundance of *Helicobacter* spp. in mice which eventually developed CRC than those which did not (Fig 4A). Therefore, the degree of *Helicobacter* spp. colonization can be considered predictive of CRC development only in SFB- mice. The diminished *Helicobacter* spp. colonization in SFB+ mice suggests that SFB provides colonization resistance against *Helicobacter* spp. and in the context of SFB, *Helicobacter* spp. colonization may be less important for the development of CRC. When further separated by sex, no differences in *Helicobacter* spp. kinetics between male and female mice were seen (S6 Fig).

**Table 2. Spearman's ranked correlation of bacterial OTUs at Pre, D4, and D7 in SFB+ and SFB- mice.**

| Timepoint | GM | OTU | R value | P value |
|---|---|---|---|---|
| **Pre** | **SFB+** | **Bilophila unc. Bacterium** | **-0.643** | **0.000218** |
| **Pre** | **SFB+** | **Lachnospiraceae unculture unc.** | **-0.615** | **0.000522** |
| **Pre** | **SFB+** | **Prevotella 9 unc. Bacterium** | **-0.606** | **0.000664** |
| **Pre** | **SFB+** | **Ruminiclostridium 5 unc.** | **-0.508** | **0.006** |
| **Pre** | **SFB+** | **Ruminiclostridium 9 unc.** | **-0.508** | **0.006** |
| Pre | SFB+ | Lachnospiraceae unc. Taxa | -0.426 | 0.024 |
| Pre | SFB+ | Ruminococcaceae UCG-003 unc. | -0.41 | 0.0303 |
| Pre | SFB+ | [Eubacterium] xylanophilum group | -0.399 | 0.0357 |
| Pre | SFB+ | Lachnospiraceae NC2004 group unc. | -0.392 | 0.0394 |
| Pre | SFB+ | Parasutterella Amb taxa | 0.484 | 0.00924 |
| **D4** | **SFB+** | **Prevotella 9 unc.Bacterium** | **-0.725** | **0.00000002** |
| **D4** | **SFB+** | **Alistipes unc. Bacterum** | **-0.555** | **0.00232** |
| **D4** | **SFB+** | **Lachnospiraceae unc. taxa** | **-0.536** | **0.00345** |
| D4 | SFB+ | GCA090006657 unc. Bacterium | -0.443 | 0.0185 |
| D4 | SFB+ | Lachnospiraceae unc. Unc. | -0.434 | 0.0212 |
| D4 | SFB+ | Lachnoclostridium 5 unc. | -0.395 | 0.0377 |
| D4 | SFB+ | Helicobacter Ambiguous Taxa | 0.00721 | 0.498 |
| D4 | SFB+ | Akkermansia unc. Bacterium | 0.385 | 0.0431 |
| D4 | SFB+ | Parasutterella Amb. Taxa | 0.4 | 0.0349 |
| D4 | SFB+ | Blautia unc. Bacterium | 0.434 | 0.0212 |
| D4 | SFB+ | Parasutterella unc. bacterium | 0.482 | 0.00969 |
| **D4** | **SFB+** | **Faecalibaculum unc. Bacterium** | **0.592** | **0.000951** |
| **D7** | **SFB+** | **Prevotella 9 unc. Bacterium** | **-0.641** | **0.00018** |
| **D7** | **SFB+** | **Lachnospiraceae A2 unc.** | **-0.625** | **0.000296** |
| **D7** | **SFB+** | **incertae sedis unc. Bacterium** | **-0.592** | **0.000769** |
| **D7** | **SFB+** | **Lachnospiraceae unc. Unc.** | **-0.581** | **0.00102** |
| **D7** | **SFB+** | **Ruminiclostridium 9 unc.** | **-0.569** | **0.00138** |
| **D7** | **SFB+** | **Lachnospiraceae UCG-006 unc.** | **-0.568** | **0.00139** |
| **D7** | **SFB+** | **GCA090006657 unc. Bacterium** | **-0.558** | **0.00176** |
| **D7** | **SFB+** | **Lachnospiraceae NC2004 group unc.** | **-0.55** | **0.0021** |
| **D7** | **SFB+** | **Lachnospiraceae 28–4 Unc.** | **-0.534** | **0.003** |
| **D7** | **SFB+** | **Ruminococcaceae unc. Unc.** | **-0.52** | **0.004** |
| **D7** | **SFB+** | **[Eubacterium] xylanophilum group** | **-0.514** | **0.00459** |
| **D7** | **SFB+** | **Lachnospiraceae NK4136 group unc.** | **-0.512** | **0.00469** |
| **D7** | **SFB+** | **Bilophila unc. Bacterium** | **-0.501** | **0.00579** |
| D7 | SFB+ | Lachnospiraceae unc. Taxa | -0.478 | 0.00892 |
| D7 | SFB+ | Ruminicoccus 1 unc. Bacterium | -0.478 | 0.00892 |
| D7 | SFB+ | Rhodospirillales unc. Bacterium | -0.456 | 0.0132 |
| D7 | SFB+ | Lachnospiraceae NK4A136 group | -0.446 | 0.0155 |
| D7 | SFB+ | Lachnoclostridium unc. Bacterium | -0.44 | 0.0173 |
| D7 | SFB+ | Ruminiclostridium unc. Bacterium | -0.43 | 0.0199 |
| D7 | SFB+ | Intestinimonas unc. Bacterium | -0.429 | 0.0204 |
| D7 | SFB+ | Oscillibacter unc. Bacterium | -0.411 | 0.0268 |
| D7 | SFB+ | Ruminococcaceae UCG-003 unc. | -0.389 | 0.0372 |
| D7 | SFB+ | Ruminiclostridium 5 unc. | -0.386 | 0.0388 |
| D7 | SFB+ | Escherichia-Shigella unc. | 0.412 | 0.0264 |
| D7 | SFB+ | Parasutterella amb. taxa | 0.496 | 0.00643 |

*(Continued)*

**Table 2.** (Continued)

| Timepoint | GM | OTU | R value | P value |
|---|---|---|---|---|
| **D7** | **SFB+** | **Helicobacter Ambiguous Taxa** | **0.536** | **0.00284** |
| **D7** | **SFB+** | **Akkermansia unc. Bacterium** | **0.55** | **0.00212** |
| Pre | SFB- | Blautia unc. Bacterium | -0.413 | 0.0092 |
| Pre | SFB- | Enterococcus Amb. Taxa | -0.363 | 0.0232 |
| Pre | SFB- | Parasutterella Amb. Taxa | -0.359 | 0.025 |
| Pre | SFB- | Parasutterella unc. Bacterium | -0.356 | 0.0261 |
| Pre | SFB- | Odoribacter unc. Bacterium | -0.341 | 0.034 |
| Pre | SFB- | Roseburia Amb. Taxa | 0.331 | 0.0396 |
| Pre | SFB- | Enterorhabdus unc. Bacterium | 0.336 | 0.0368 |
| Pre | SFB- | Lachnospiraceae unc. unc. | 0.339 | 0.0347 |
| Pre | SFB- | GCA090006657 unc. Bacterium | 0.344 | 0.0323 |
| Pre | SFB- | Intestinimonas unc. Bacterium | 0.345 | 0.0316 |
| Pre | SFB- | Ruminiclostridium 9 unc. | 0.35 | 0.0293 |
| Pre | SFB- | [Eubacterium] corpostanoligene | 0.354 | 0.0274 |
| Pre | SFB- | Lachnospiraceae 28–4 Unc. | 0.355 | 0.0269 |
| Pre | SFB- | Bacteroides unc. Bacterium | 0.36 | 0.0244 |
| Pre | SFB- | Lachnospiraceae unc. taxa | 0.366 | 0.022 |
| Pre | SFB- | Ruminiclostridium 5 unc. | 0.373 | 0.0195 |
| Pre | SFB- | Lachnoclostridium unc. Bacterium | 0.387 | 0.0153 |
| Pre | SFB- | Ruminococcus 1 unc. Bacterium | 0.411 | 0.00954 |
| Pre | SFB- | Oscillibacter unc. Bacterium | 0.416 | 0.00865 |
| Pre | SFB- | Ruminococcaceae UCG-003 unc. | 0.451 | 0.00418 |
| Pre | SFB- | Ruminococcaceae unc. Unc. | 0.452 | 0.00404 |
| Pre | SFB- | Lachnospiraceae A2 unc | 0.469 | 0.00276 |
| D4 | SFB- | Akkermansia unc. Bacterium | -0.375 | 0.00971 |
| D4 | SFB- | Ruminiclostridium unc. Bacterium | -0.351 | 0.0159 |
| D4 | SFB- | Lachnospiraceae UCG-001 unc. | -0.31 | 0.0339 |
| D4 | SFB- | Lachnospiraceae A2 unc. | 0.325 | 0.0258 |
| D4 | SFB- | Ruminococcaceae UCG-003 unc. | 0.34 | 0.0196 |
| D4 | SFB- | Helicobacter unc. Bacterium | 0.431 | 0.00263 |
| D7 | SFB- | Akkermansia unc. Bacterium | -0.34 | 0.017 |
| D7 | SFB- | Lachnospiraceae UCG-008 unc. Bacterium | 0.312 | 0.0295 |
| D7 | SFB- | Roseburia Amb. Taxa | 0.32 | 0.025 |
| D7 | SFB- | Fusobacterium unc. Bacterium | 0.321 | 0.0249 |
| D7 | SFB- | Lachnospiraceae A2 unc. Bacterium | 0.337 | 0.00779 |
| D7 | SFB- | Helicobacter Unc. bacterium | 0.412 | 0.0034 |

Bacterial OTUs which are significantly correlated with high or low overall disease scores are listed for Pre, D4, and D7 in both SFB- and SFB+ mice. R values and P values are listed side by side. R values range from -1 to +1, with strong negative and positive correlations (respectively), closer to ±1 than 0. Bolded values indicate stronger correlations.

## Family *Enterobacteriaceae* is predictive of CRC development in SFB+ mice only

Lastly, an OTU annotated to the family *Enterobacteriaceae*, more specifically resolved as *Escherichia-Shigella* spp., demonstrated a concurrent bloom with *Helicobacter* spp. (Fig 4B). From Pre to 3W PI, SFB- mice revealed similar family *Enterobacteriaceae* kinetics regardless of CRC development, in which family *Enterobacteriaceae* bloomed at D4 and remained relatively

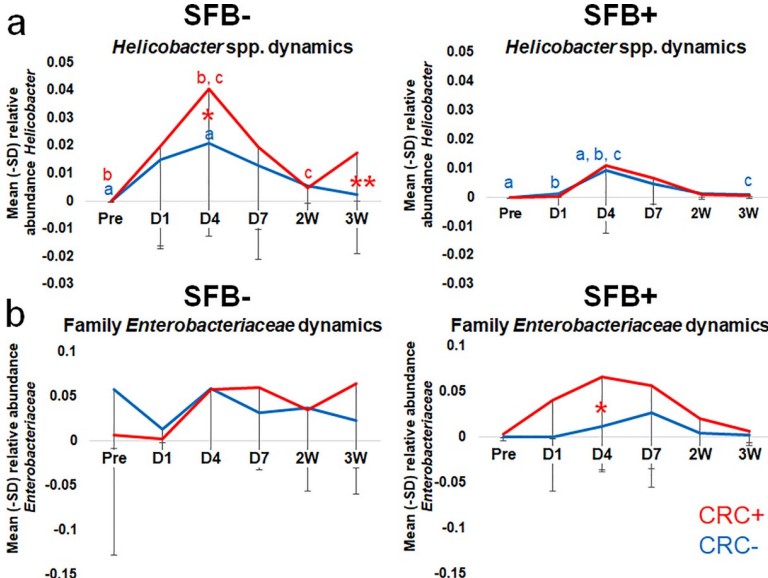

**Fig 4. Relative abundance of *Helicobacter* spp. at D4 PI predictive of CRC development in SFB- but not SFB + mice.** Family *Enterobacteriaceae* and SFB interact in CRC development. Relative abundance of (A) *Helicobacter* spp. and (B) family *Enterobacteriaceae* (*Escherichia-Shigella Escherichia* sp.) from Pre-inoculation to 3W PI in SFB- and SFB+ *Helicobacter* spp.-inoculated mice. Two-way RM ANOVA performed for SFB- and SFB+ mice separately: like letters denote significant ($p < 0.05$) differences. P values are as follows: A) SFB- group *a* $p = 0.042$; *b* $p = 0.005$; *c* $p = 0.019$; *$p = 0.016$; **$p = 0.024$ and SFB+ group *a* $p = 0.012$; *b* $p = 0.046$; *c* $p = 0.031$. B) SFB- group *ns* and SFB + group *$p = 0.048$. SFB- group CRC+ (n = 19) and CRC- (n = 38) and SFB+ group CRC+ (n = 7) and CRC- (n = 22).

stable over time. However, in SFB+ mice only, family *Enterobacteriaceae* were significantly (p = 0.048, Two-Way RM ANOVA) more abundant at D4 in mice which eventually developed CRC. This difference in relative abundance in CRC- and CRC+ SFB+ mice only could not only serve as a predictive biomarker, but hint at an interaction between SFB and family *Enterobacteriaceae* in CRC development, which should be explored further to gauge whether Family *Enterobactericeae* is interacting with SFB to drive carcinogenesis, or if this family is simply being suppressed in the SFB-CRC- group. Interestingly, when divided by sex in both SFB- and SFB+ groups, female mice demonstrated a greater relative abundance of family *Enterobacteriaceae* compared to male mice. Statistics were not performed on these data, as there was only one CRC+ male in the SFB+ group (S6B Fig). Linear discriminant analysis effect size (LEfSe) was used to search for biomarkers among groups on Pre and D4 (S7 Fig). Family *Enterobacteriaceae* relative abundance pulled out as significant within the SFB-CRC- group pre-inoculation, but then was significant within the SFB-CRC+ group at D4 PI. This is puzzling, as the average relative abundance of Family *Enterobacteriaceae* at D4 within the SFB- group is nearly identical.

Because of this relationship between family *Enterobacteriaceae* and SFB, we set out to more precisely identify the species contributing to CRC development. Fecal slurries from samples collected D4 PI were streaked onto blood agar plates (BAP) and MacConkey Agar Plates (MAC) in an anaerobic hood. Colonies were identified using matrix-assisted laser desorption/ ionization time-of-flight (MALDI-ToF) mass spectrometry. Seven of the eight samples were identified as *Escherichia coli*, as expected (Fig 5A). One strain of *E. coli* was isolated and preserved, then restreaked in comparison to ATCC strain 21972. On BAP, the clinical sample was *alpha*-hemolytic compared to the lab strain, which was *beta*-hemolytic (Fig 5B). Properties such as these could yield insight to strain-dependent differences in function of *E. coli* that aren't readily discernable from 16S sequencing data alone.

a

MALDI-TOF identification

| Isolate | #1 | #2 | #3 | #4 | #5 | #6 | #7 | #8 |
|---|---|---|---|---|---|---|---|---|
| *Clostridium sp.* | - | - | - | - | + | - | - | - |
| *Enterococcus gallinarum* | + | - | - | - | - | - | - | + |
| *Escherichia coli* | + | - | + | + | + | + | + | + |
| *Lactobacillus murinus* | + | - | - | - | - | + | - | + |
| *Parabacteroides goldsteinii* | - | - | - | - | + | - | + | + |

b

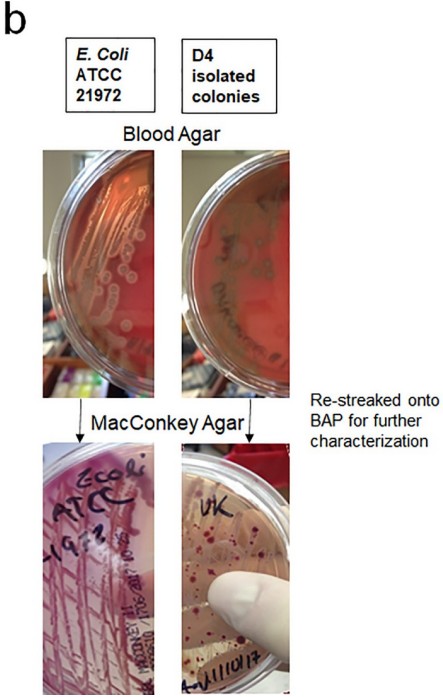

**Fig 5. Isolation of family *Enterobacteriaceae*.** A) MALDI-ToF identification of D4 PI fecal samples. B) Blood Agar and MacConkey Agar plates comparing colony morphology, hemolytic properties, and lactose fermentation of lab strain ATCC 21972 *E. coli* and unknown D4 isolated *E.coli* strain.

Illumina NextSeq technology was used to sequence these isolates for genome assembly and further analysis for potential virulence factors which may contribute to CRC development. One such target is colibactin, an enzyme already implicated in CRC development due to its ability to cause double-stranded DNA breaks in host epithelial cells. [29] This protein is encoded by the 54 kb biosynthetic gene cluster polyketide synthase (*pks*) pathogenicity island. [29] These analyses are ongoing and will include other virulence factors, but attempts to map either isolate to the *colibactin*-encoding *pks* pathogenicity island was unsuccessful.

## Discussion

Although colitis-associated CRC (CAC) accounts for only 1–2% of the total cases of CRC, it results in the death of 15% of IBD patients [5]. This risk increases with the duration of IBD symptoms and severity of inflammation—and due to CAC tumor morphology (flat and multi-focal) and aggressive histology (mucinous adenocarcinomas and signet ring carcinomas) [30], early detection can be challenging. This results in a worse prognosis and higher mortality rate in CACs than sporadic CRCs [5, 6].

Screening programs including frequent surveillance colonoscopies are invasive and don't address the underlying cause of CAC [31]. An understanding of inciting events early or prior to CAC development is necessary in order to develop preventative strategies and therapeutics to decrease the risk of CRC for IBD-sufferers. Therefore, we conducted a longitudinal study of the gut microbiota in the Smad3$^{-/-}$ mouse CAC model, focusing on early time-points to better understand the complex relationship of the gut microbiota, chronic inflammation, and tumor development.

We compared the static and dynamic differences of GMs with and without SFB through the course of CRC development. Recently at the forefront of GM studies, SFB's claim to fame revolves around its immunomodulatory properties, particularly in the induction of Th17-based immunity [16, 32], and its role in modulating the phenotype of models of immune-mediated diseases such as rheumatoid arthritis [19], type 1 diabetes (T1D) [20], and experimental autoimmune encephalomyelitis (EAE) [21]. In a mouse model of colitis, the reduction of SFB and thus Th17 pathway cytokines by early exposure to penicillin was correlated to a reduction in colitis severity [33].

SFB has also been detected in humans, most prominently in children three years of age or younger, as levels decrease below the limit of detection by adulthood [23]. This human variant of SFB results in higher titers of IgA and an up-regulation of Th17 pathways in humans, as in mice [23]. Interestingly, prior to this study, one anecdotal report suggested high numbers (>50 filaments) of SFB-like organisms in histological slides of ileo-cecal valves of patients suffering from UC [34]. Only half of healthy controls contained SFB and in numbers of five filaments or less [34]. Finotti *et al* also detected SFB via PCR in the terminal ileum of patients with UC [34]. Thus, our hypothesis was that SFB-mediated Th17 responses would exacerbate colitis and the development of CAC.

In the case of the Smad3$^{-/-}$ CAC model, SFB did not affect overall disease incidence and severity, which was surprising because Th17 levels have been associated with poor CRC prognosis in past studies [35]. However, it impacts the phenotype of this model in more subtle ways. For example, SFB plays a role in stabilization of the GM despite the introduction of inflammatory provocateurs *Helicobacter hepaticus* and *H. bilis*, and appears to play a role in colonization resistance despite different intestinal niches. Similar instances of SFB-dependent colonization resistance have been seen in cases of *Citrobacter rodentium* [16, 36]. Relative abundance of *Helicobacter* spp. at D4 PI in SFB- mice is predictive of CRC development, but the mouse model relies less heavily on the colonization of *Helicobacter* spp. trigger if SFB is present. Most intriguingly, only in the presence of SFB does family *Enterobacteriaceae*, most notably *E. coli* strains, predict CRC development, while SFB- mice exhibit no diverging pattern of *E. coli* regardless of eventual CRC status. These data suggest a possible interaction between family *Enterobacteriaceae* and SFB in disease development and decreased reliance on *Helicobacter* spp.-colonization than in SFB- mice. The exact role *E. coli* plays is unclear, though there is evidence in other models of virulence factors such as *colibactin* contributing to tumor development through double-stranded DNA breakage [37]. The isolation of an α-hemolytic strain of *E.coli* at D4 PI is intriguing, as invasive species of *E. coli*, rather than facultative strains, tend to produce virulence factors responsible for α-hemolysis [38], and are often isolated from extraintestinal infections [39] and intraperitoneal infections following breach in intestinal barriers [40]. Thus, we propose a model wherein the classic "driver and passenger" model of bacterial community dynamics during CAC, wherein the reliance of disease changes from the pathobiont "driver" (*Helicobacter* sp.) to a pathobiont "passenger" (*E. coli*), depending on the presence or absence of a third bystander (SFB) and its associated colonization resistance against the "driver".

Thus, while SFB does not modulate the overall incidence or severity of CAC in *Helicobacter* spp.-inoculated Smad3$^{-/-}$ mice, we hypothesize that it changes the underlying mechanism of

disease development. These mice rely less on *Helicobacter* spp.- induced dysbiosis than their SFB- counterparts—and may even bring about colonization resistance against this and other mucosa-adherent or invasive bacteria. Because SFB is instrumental in the development of a Th17 compartment in the small intestine lamina propria [16], we propose that the SFB disease mechanism relies more so on an autoimmune Th17 response. Th17 cells produced under conditions of IL-1β and IL-6 exposure with little to no canonical TGF-β signaling (which relies on Smad3 signaling) or with non-canonical TGF-β signaling (which bypasses Smad3) have been reported as pathogenic in EAE models [41, 42]. The lack of Smad3 signaling in this model and the Th17-inducing properties of SFB support this.

Also intriguing are the apparent sex-dependent effects of SFB in the development of CRC. While the greater incidence of CRC in female mice relative to male mice in the absence of SFB did not achieve statistical significance, the sex bias in disease incidence achieved significance in the presence of SFB. Specifically, we speculate that SFB confers a selective protection to male mice, which have reduced disease incidence compared to all other groups. To our knowledge, only one other study has reported a sex-dependent modulation of phenotype by SFB. In 2011, Kriegal *et al* documented a discordant penetrance of diabetes in NOD mice in their facility in comparison to reports from Jackson Laboratories [20]. However, they found that SFB afforded female mice the same protection from disease development that both SFB+ and SFB-males already benefited from [20].

Several population-based cohort studies found that males were more likely to develop CRC following a diagnosis of IBD than females [43, 44], though IBD and autoimmune diseases are disproportionately skewed toward females [45]. These data, however, do not explain why only male mice are protected by SFB in the Smad3$^{-/-}$ model. Several studies, however, look to androgen treatment for the down-regulation of inflammation in autoimmune and inflammatory models. Traish *et al* found that androgen deficiency in men was correlated with higher levels of pro-inflammatory cytokines such as IL-6, IL-1β, and TNF-α [46]. Another study found that the gut microbiota from adult male NOD mice conferred protective effects to weanling female NOD mice which was not vertically transmissible to offspring [47]. When investigated further, the male microbiome increased the serum levels of testosterone in those females [47]. Castration of NOD males increased diabetes incidence [48], while exogenous androgen therapies decreased incidence in females [49]. Taking it one step further, Yurkovetskiy documented the ability of SFB to enhance testosterone production and protection against disease incidence selectively in male mice in a model of type-I diabetes [50]. Thus, our current working hypothesis as to how SFB confers protection to only male mice is predicated on a link between testosterone-related changes in gene expression, including decreases in IL-6, a cytokine required for Th17 T cell differentiation, and the male-restricted protection in SFB+ mice. It is possible that SFB is stabilizing the GM and protecting the mucosa from *Helicobacter* spp.-colonization, yet bringing about CRC through a Th17-driven means, which is abrogated by higher levels of testosterone in male mice. More work must be done to investigate Th17 cytokines in male and female SFB+ mice in the context of the Smad3$^{-/-}$ model.

In conclusion, these data emphasize the complexity of IBD-driven CRC development, as well as how key members of the GM may subtly alter a mouse model without obvious changes to disease incidence. Because SFB colonization varies between mouse suppliers [16, 17, 20], this threatens the reproducibility of studies and calls for consideration of how microbial variables, both dependent and independent, may act in a variety of contexts. As a matter of translatability, this could also impact how well certain treatments or preventative strategies work in different people in different environments. As a consideration in the design of precision medicine approaches to CAC, the goal of future studies will be to better understand the mechanistic relationship between major players of the GM, inflammatory bowel diseases, and CRC development.

## Materials and methods

### Animals

Smad3$^{-/-}$ (129-Smad3$^{tmPar}$/J) mice, originally a generous gift from Lillian Maggio-Price, were bred on-site and group-housed in microisolator cages on ventilated racks and provided auto-claved food and water (acidified) *ad libitum*. Multiple cohorts of mice were used as they were ready, combined into groups based on birthdate, such that inoculation with *Helicobacter* spp. could occur at 3 weeks of age (weaning). Segmented filamentous bacteria (SFB) groups were generated using breeding pairs colonized with the same GM as the main colony, but that were experimentally inoculated with SFB in order to generate SFB+ offspring. Mice were aged to 14 weeks post-inoculation before euthanasia by $CO_2$ asphyxiation and secondary cervical disloca-tion. All animal procedures were approved by the University of Missouri IACUC and per-formed in accordance with the Guide for the Care and Use of Laboratory Animals, and AVMA Guidelines on Euthanasia.

### SFB PCR confirmation

PCR was performed on fecal samples at Pre, D1 and D4 post-inoculation with *Helicobacter* spp. to confirm presence of SFB. Mice within the SFB+ cohort which tested negative at all of these timepoints were removed from the study. SFB- groups were also tested at random, to confirm the absence of SFB. SFB779 forward (IDT ref: 215274473) and SFB1008 reverse (IDT, ref: 215274472) primers were used in a 25 μL/sample reaction containing 12.875 μL nuclease-free water (Promega, cat: p1193), 2.50 μL 10× PCR buffer + $MgCl_2$ (Roche, cat: 12161516103), 4 μL 5 mM dNTPs (Roche, cat: 11969064001), 1.25 μL SFB779 forward, 1.25 μL SFB1008 reverse, [51, 52] and 0.125 μL Faststart Taq DNA Polymerase (Roche, cat: 12032953001) mixed with 3 μL DNA. The following parameters were used: 1 cycle of lysis for 3 minutes at 95˚C and 40 cycles of denaturation for 15 seconds at 95˚C, annealing for 30 seconds at 59˚C, and extension for 30 seconds at 72˚C.

### Bacterial culture

For *Helicobacter hepaticus*, three blood agar plates per cohort were prepared with 6.0 mL Bru-cella broth (BD Difco$^{tm}$ BBL$^{tm}$, cat: BD211088) and inoculated with equal portions of a 1 mL frozen aliquot of glycerol (10%)-preserved *H. hepaticus*. Inoculated plates were incubated at 37˚C at a slight tilt in bell jars flushed with $CO_2$ for 45 seconds and maintained at a pressure of 5 PSI overnight. Upon inspection for purity and robustness under an inverted light micro-scope, plates of *H. hepaticus* were passaged into a single flask containing a stir bar and 10% fetal bovine serum (FBS) in Brucella broth, equal to 35 mL total. Flasks were added to a bell jar, flushed as before, and maintained at 5 PSI pressure overnight at 37˚C. The final passage was inspected under the inverted microscope for purity and robust growth of bacteria.

For *Helicobacter bilis*, 1 mL of frozen glycerol stock was transferred to a flask containing 35 mL of Brucella broth supplemented with 10% FBS to, as described above. The bell jars were flushed similarly with $CO_2$, held at 5 PSI pressure, and incubated at 37˚C overnight. Cultures were also inspected via inverted microscope for purity and robust growth of motile bacteria.

### *Helicobacter* spp. inoculation

Pure cultures of *Helicobacter hepaticus* and *H. bilis* were combined equally in a 50 mL conical and inverted to mix. Using a curved needle with a ball-tip (i.e., gavage needle), weanling Smad3$^{-/-}$ mice were administered 0.5 mL of ~$10^8$ *Helicobacter* mixture each by gastric gavage.

Sham-inoculated mice are given 0.5 mL of Brucella broth. Each mouse received two inoculations, 24 hours apart.

## Fecal sample collection

Mice were placed in individual autoclaved cages absent of substrate and allowed to defecate naturally. Using sterile toothpicks, freshly evacuated fecal pellets were immediately placed into 800 μL of lysis buffer in a 2.0 mL round-bottom tube containing a sterile 0.5 cm-diameter stainless steel ball bearing. To monitor the GM over time, fecal samples were collected prior to the first inoculation (pre), prior to the second inoculation (mid), then one day (D1), D4, D7, two weeks (2W), 3W, 5W, 8W post-inoculation (PI), then at sacrifice (14W).

## Tissue sample collection

Following euthanasia at 14W, mice were necropsied and their colon, cecum, and ileum removed as one continuous piece. The most aboral fecal pellet was removed from each mouse and added to a round bottomed tube as described before. Cecal contents were collected similarly, using flame sterilized scissors to cut open the cecum and a sterile toothpick to gather up the contents. The tissue was flushed with saline then fixed in 10% formalin. The entire colon was embedded in paraffin and 5 μm-thick longitudinal sections were prepared and stained with hematoxylin and eosin for histological examination and lesion scoring.

## DNA extraction

DNA extraction was performed using a two-stage process first described by Yu et al [53] and adapted for murine samples by Ericsson *et al* [2]. Briefly, fecal DNA was extracted using an ammonium acetate/isopropanol protocol first described by Yu and Morrison [53]. The pellet was resuspended in Tris-EDTA buffer and purified over a DNeasy column using the manufacturer's instructions for DNA extraction from cells using the Qiagen DNeasy Blood and Tissue Kit (Qiagen, cat: 69506). The DNA was eluted in EB Buffer rather than the supplied AE buffer. Final DNA concentrations were quantified using a Qubit 2.0 fluorometer and Qubit dsDNA BR Assay Kit (Invitrogen, cat:Q32853).

## 16S rDNA library preparation and sequencing

Library construction and sequencing was performed in a 96-well multiplexed format by the University of Missouri DNA Core as described previously [14, 54]. In brief, dual-indexed universal primers F515/R806, flanked by Illumina adaptor sequences, were used to construct amplicons of the V4 hypervariable regions of 16s rRNA gene. Amplicon libraries were pooled and sequenced on the Illumina MiSeq platform and the V2 chemistry [55].

## Informatics analysis

Informatics were performed by the University of Missouri Informatics Research Core Facility. FLASH (Fast Length Adjustment of SHort reads) software was used to merge DNA sequences [56]. Reads were truncated if the base quality was less than 31 and removed if total length was less than the expected 292 basepairs [54, 55]. Primers on both ends of the contigs were removed with cutadapt, [57] and sequences initially lacking primers on one or both ends were deleted. Contigs with expected errors <0.5 were retained and trimmed for quality using the USEARCH fastq_filter command then clipped to 248 bases each [58]. Samples were de-multiplexed using Qiime v1.9 (split_libraries_fastq.py) and concatenated into a single file [59]. Samples were clustered into OTUs, based on 97% similarity cut-off using the uparse method [60],

then assigned taxonomy using BLAST against SILVA database v132 (http://www.arb-silva.de) of 16S rRNA sequencing and taxonomy [61].

## Histopathology

H&E-stained slides were analyzed by a trained veterinary pathologist, blinded to treatment groups, and given colonic lesion scores based on epithelial changes, inflammation, and tumor size and invasiveness as follows:

Epithelial Changes: Hyperplasia/Dysplasia (1–5), Longitudinal Extent (0–4), Total Epithelial Score (Hyperplasia/Dysplasia Score × Longitudinal Extent Score).

Inflammation: Inflammatory Severity (0–3), Longitudinal Extent (0–4), and Total Inflammatory Score (Inflammatory Severity × Longitudinal Extent).

Tumor Score: Tumor Size (#10× fields/tumor), Invasiveness (1–3), and Total Tumor Score (Tumor Size × Invasiveness)

Overall Disease Score: Total Epithelial Changes Score + Total Inflammation Score + Total Tumor Score

## Anaerobic bacterial culture and identification

Day 4 PI fecal samples were collected from SFB+ and SFB- *Helicobacter*-inoculated mice into 800 μL sterile water in a sterile 2 mL round-bottom tube. Samples were lysed via TissueLyser and taken into the anaerobic hood. Each sample was streaked onto Blood Agar (BAP) and MacConkey Agar (MAC) plates. The two distinct morphologies isolated on MAC plates were re-streaked for additional purity and frozen back as stock in 10% glycerol. The BAP were examined via matrix-assisted laser desorption-ionization time-of-flight (MALDI-ToF) mass spectrometry (Bruker Microflex LT MALDI-TOF Mass Spectrometer) for strain identification (IDEXX BioAnalytics) using the Bruker Daltonics Database (BDAL).

## *E.coli* Isolate genome sequencing

The two separate *E. coli* isolates were submitted to the MU DNA Core for sequencing on the Illumina NextSeq. The University of Missouri Informatics Research Core Facility mapped isolate reads against AM229678 EMBL database *pks* sequence data using bowtie2 (version 2.3.4.3). Samtools (version 0.1.19-44428cd) was used to create a table of read depth for each position in the *pks* reference, which was compared visually with data found at https://www.ebi.ac.uk/ena/data/view/AM229678&display=text to identify annotated regions of the *pks* reference.

## Statistical analysis

Histological scoring was performed as described above and paired with the longitudinal GM data for each mouse. Incidence was determined based on presence/absence of tumor in the colon and categorized by SFB status then further divided by sex within each category. Chi-square and Chi-square further delineated using individual Fisher's Exact tests respectively, measured significance along those two groupings. Tumor scores, inflammation scores, and overall disease scores were similarly separated by SFB status and then further divided by sex within these categories. Kruskal-Wallis one-way ANOVA on ranks and Two-Way ANOVA tests (*post hoc* Holm-Sidak) were used respectively to determine statistical differences between groups.

Multivariate analysis including generation of Principal Coordinate Analysis (PCoA), loading plots, and PERMANOVA statistical analysis were performed using PAleontological

STatistics (PAST) [28]. Separate PCoAs, using Bray-Curtis and Jaccard diversity indices, were generated to compare CRC+ and CRC- animals in SFB+ and SFB- groups at various time-points, including pre-inoculation and D4 post-inoculation.

Overall GM community shifts between corresponding Pre and D4 samples of SFB+ and SFB- mice were analyzed using the Bray-Curtis dissimilarity Index using PAST [28]. The Mann-Whitney Rank Sum test determined statistical significance between groups. These groups were further divided by sex within SFB group and analyzed using a Two-Way ANOVA (*post hoc* Holm-Sidak).

Bacterial kinetics of *Helicobacter* spp. (both *Helicobacter* OTUs detected were averaged together into one value per sample) and family *Enterobacteriaceae* were generated by taking the averages of relative abundances of each OTU from Pre to 3W post-inoculation timepoints, using Microsoft Excel for SFB- and SFB+ groups. Two-way Repeated Measures ANOVA (*post hoc* Bonferroni), performed separately for SFB- and SFB+ groups, determined statistical significance and interactions between CRC development, timeline, and relative abundance of the target OTU. Kinetics were further divided on basis of sex, but stats were not performed as animal numbers per group were too low.

Linear discriminant analysis effect size (LEfSE) [62] was performed at the OTU level for D1 and D4 for SFB-CRC-, SFB-CRC+, SFB+CRC-, and SFB+CRC+ groups. The lowest number of counts per OTU was set at 12.

Spearman rank correlations were performed using SigmaPlot version 14.0. In short, SFB+ and SFB- correlations were performed separately for Pre, D4, and D7 timepoints at the OTU and Family level, using ranked overall scores. The R values for OTUs and families with a p value <0.05 were recorded.

To account for potential clustering effects of multiple animals being housed in the same cage, linear mixed effects models generated by the lme4 package were used with cage number as a random effect [63]. Models with sex and SFB status as independent variables were evaluated, as well as models with multiplicative interactions between sex and SFB status. The Car package used to generate P values from lme4-generated mixed modeling data, using the Anova function [64].

## Supporting information

**S1 Fig. SFB does not impact overall penetrance or severity of CAC.** Penetrance of CRC a), Tumor Scores b), and Overall Disease Scores c) in SFB- and SFB+ groups. Chi Square a) and Kruskal-Wallis One Way ANOVA on Ranks (b and c) performed.
(TIF)

**S2 Fig. No early GM profile found to be predictive of CRC development.** Principal Component Analysis (PCoA) graphs utilizing the a) Bray-Curtis and b) Jaccard diversity indices of pre-inoculation timepoints for SFB- and SFB+ cohorts which did and did not develop CRC. PERMANOVA (permutation number = 9999) on each PCoA between CRC+ and CRC- groups. A) SFB- Bray-Curtis p = 0.1626; SFB+ Bray-Curtis p = 0.5162 B) SFB- Jaccard p = 0.1398; SFB+ Jaccard p = 0.1161.
(TIF)

**S3 Fig. Examining the effects of cage and sex on CRC.** PCoAs using Bray-Curtis indices of a) CRC and SFB and Sex and SFB categorized pre-inoculation samples and b) SFB- and SFB+ mice categorized by cage. One-way PERMANOVA determined statistical differences in A between SFB+CRC+ mice and SFB- mice regardless of CRC fate (SFB-CRC- p = 0.0006; SFB-CRC+ p = 0.0204). Using Bonferroni-corrected p values, with a permutation number of

9999, Male SFB+ mice were statistically different from both male (p = 0.0114) and female (p = 0.0006) SFB- mice but not female SFB+ mice (p = 0.1758) at Pre-inoculation timepoint. Pairwise comparisons included below each PCoA.
(TIF)

**S4 Fig. Accompanying figure legend to Fig 3A.** Corresponding OTUs to each segment of the barcharts shown on Fig 3A.
(TIF)

**S5 Fig. Bray-Curtis dissimilarity Index examining total GM shifts between female and male mice within the SFB- and SFB+ groups from Pre-inoculation to D4 PI.** Two-Way Anova (*post hoc* Holm-Sidak) test performed. Only the overall differences between SFB+ and SFB- was significant (p = 0.015).
(TIF)

**S6 Fig. *Helicobacter* spp. and Family *Enterobacteriaceae* kinetics between male and female mice.** Relative abundance of a) *Helicobacter* spp. and b) Family *Enterobacteriaceae* in male and female SFB- and SFB+ mice which did and did not develop CRC. No stats were performed due to a low n.
(TIF)

**S7 Fig. LEfSe analysis at early timepoints.** Linear discriminant analysis effect size (LEfSe) on a) Pre-inoculation and b) D4 post-inoculation in SFB+CRC-, SFB+CRC+, SFB-CRC+, and SFB-CRC- groups.
(TIF)

## Acknowledgments

We would like to acknowledge the MU DNA Core and MU Informatics Research Core facilities for their invaluable services in sequencing and analyzing the 16S rRNA data.

## Author Contributions

**Conceptualization:** Annie E. Wolfe, Aaron C. Ericsson.

**Data curation:** Aaron C. Ericsson.

**Formal analysis:** Annie E. Wolfe, Jacob E. Moskowitz, Timothy L. Wiemken.

**Funding acquisition:** Craig L. Franklin, Aaron C. Ericsson.

**Investigation:** Annie E. Wolfe, Jacob E. Moskowitz, Aaron C. Ericsson.

**Methodology:** Aaron C. Ericsson.

**Resources:** Aaron C. Ericsson.

**Supervision:** Craig L. Franklin, Aaron C. Ericsson.

**Writing – original draft:** Annie E. Wolfe.

**Writing – review & editing:** Annie E. Wolfe, Craig L. Franklin, Aaron C. Ericsson.

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
