## [Decision Letter · Decision Letter 0]

22 Apr 2020

PONE-D-20-08853

Interactions of Segmented Filamentous Bacteria (Candidatus Savagella) and bacterial drivers in colitis-associated colorectal cancer development.

PLOS ONE

Dear Dr. Ericsson

Thank you for submitting your manuscript to PLOS ONE. After careful consideration, we feel that it has merit but does not fully meet PLOS ONE’s publication criteria as it currently stands. Therefore, we invite you to submit a revised version of the manuscript that addresses the points raised during the review process.

Both reviewers feel that the study is potentially interesting, however, they also feel that the results presented here are not sufficient to support the authors' conclusion.  Moreover, one reviewer has also a serious concern of the statistical methods used in the study.

We would appreciate receiving your revised manuscript by June 21, 2020. To enhance the reproducibility of your results, we recommend that if applicable you deposit your laboratory protocols in protocols.io, where a protocol can be assigned its own identifier (DOI) such that it can be cited independently in the future. For instructions see: http://journals.plos.org/plosone/s/submission-guidelines#loc-laboratory-protocols

We look forward to receiving your revised manuscript.

Kind regards,

Hiroyasu Nakano, M.D., Ph.D.

Academic Editor

PLOS ONE

Journal Requirements:

"This work was funded in part by NIH grants K01 OD019924 and U42 OD010918. A.W. was supported by a University of Missouri Life Science Fellowship."

"Funding for this research was provided as follows:  

C.F. received funding (Grant#: U42 OD010918) from

NIH, Office of the Director  (https://www.nih.gov/institutes-nih/nih-office-director)

A.E. received funding (grant#: K01 OD019924 ) from NIH, Office of the Director  (https://www.nih.gov/institutes-nih/nih-office-director)

Reviewers' comments:

Reviewer's Responses to Questions

**Comments to the Author**

1. Is the manuscript technically sound, and do the data support the conclusions?

Reviewer #1: Partly

Reviewer #2: Partly

2. Has the statistical analysis been performed appropriately and rigorously? 

Reviewer #1: I Don't Know

Reviewer #2: No

3. Have the authors made all data underlying the findings in their manuscript fully available?

Reviewer #1: Yes

Reviewer #2: Yes

4. Is the manuscript presented in an intelligible fashion and written in standard English?

Reviewer #1: Yes

Reviewer #2: Yes

5. Review Comments to the Author

Reviewer #1: Wolfe et al. reported in this manuscript that CRC incidence, disease scores, tumor scores, and inflammation scores after Helicobacter spp. inoculation in Smad3-/- mice were significantly reduced in male mice compared to female mice only when SFB was inoculated. In addition, the authors showed that SFB inhibited colonization of Helicobacter spp. Relative abundance of Helicobacter spp. was predictive of CRC development in SFB- mice whereas E. Coli were significantly more abundant in mice which eventually developed CRC in SFB+ mice. Although the study was well planned and revealed interacting phenomena regarding intestinal bacterial colonization during CRC development in the Helicobacter spp-Smad3-/- mouse model, it was too descriptive. More rigorous examination needs to be performed. Specific comments are below.

1. Immune cell population and activation including expansion of Th17 cells and production of Th17 cytokines need to be tested in male and female mice in the presence/absence of SFB.

2. There were no differences in GM stability on the basis of sex in either SFB+ or SFB- mice (Fig. S4). Since the authors showed that CRC development was reduced in male SFB+ mice, this observation seems to contradict with the conclusion “Relative abundance of Helicobacter spp. was predictive of CRC development in SFB- mice”.

Minor comment

3. This reviewer suggests to present representative pictures of intestinal histology in addition to quantified results presented in the manuscript.

Reviewer #2: Wolfe et al., described the protective role of SFB on colorectal cancer (CRC) of smad3-/- male mice. However, it is unclear that the interactions among three bacteria, SFB, Helicobacter and E. coli are relevant to the disease severity of CRC, because key and comprehensive analyses are missing in this manuscript. More detailed comments are provided below:

Comments:

1. It is unclear what Figure 3a shows. The text says " relative abundance of operational taxonomic units (OTUs)" (Line 169), but the figure has " Enterobacteriaceae" (family) and "Helicobacter spp." (genus), implying that the analysis is at family or genus level. Which taxonomic level did the authors use, OTU@>97%, genus, or family? If it is possible, please use yellow and blue to make the highlighted taxons clearer in the panel. Moreover, description of other OTUs is missing. I would like to request the supplemental table of OTU abundance of individual samples to review and judge the results of figures in this manuscript.

2. For the conclusion "Three patterns emerged from pre-inoculation to 2W PI within the SFB-CRC-, SFB-CRC+, SFB+CRC-, and SFB+CRC+ groups. First, the GM of SFB+ mice, regardless of CRC development, remained relatively consistent from pre to 2W post inoculation when compared to the GM of SFB- mice. Second, colonization with Helicobacter spp. peaked at D4 in SFB- mice regardless of CRC development, but was blunted in SFB+ mice. Lastly, unresolved microbes in the family Enterobacteriaceae bloomed concurrently with Helicobacter spp. in all groups but SFB+CRC-." (Lines 170 - 176), the authors should perform multiple statistical analyses including LEfSe and FDR.

3. The conclusion "Subjectively, SFB- mice appear to have greater community shifts following Helicobacter spp.-inoculation than SFB+ mice. " (Lines 188 - 190) and following statement require the result of statistical analysis.

4. "On average, SFB- Pre and D4 time-points were more dissimilar than Pre and D4 time-points of SFB+ mice (p = 0.023, Mann-Whitney rank sum test)," (Line 191 to 193). The authors should show medians but not averages as the data was non-parametric as they used Mann-Whitney rank sum test. The comparison also needs controls, Bray-Curtis indexes of internal Pre and D4, because the difference might simply reflect the internal variations of Pre or D4, and it must be multiple comparison test and another test other than Mann-Whitney is required. I have a similar concern on Figure S4.

5. The analysis of the sections "Relative Abundance of Helicobacter spp. at D4 PI predictive of CRC development in SFB- but not SFB+ mice." and Family Enterobacteriaceae and SFB interact in CRC development is biased for particular bacteria. The authors should perform Spearman ranking test of all (from higher to lower) taxons with CRC levels. This is the most critical point for the science of this manuscript.

6. "Colonies were identified using matrix-assisted laser desorption/ionization time-of-flight (MALDI-ToF)

mass spectrometry. Seven of the eight samples were identified as Escherichia coli, as expected (Figure

5A)" Because Figure 5A too much simplified the result of MALDI-TOF, the authors should attach the supplemental tables of identified bacterial molecules and describe which database was used for identification of the molecules. Morerover, the authors should describe justification for the reason why they used mass spec as genome sequencing of isolates gives more accurate and detailed results with low cost.

7. According to the abundance of Enterobacteriaceae in Figures 3a and 4, the fact that the result of Figure 4a showed the dominance of E. coli, suggests cloning bias of bacterial isolates. This raises the concern about bias in the conclusion.

8. What does the finding on hemolytic features of one E. coli strain (Fig, 5b) mean? Is this associated with the conclusions of this manuscript?

Minor comments:

9. The figure legend is missing. How many mice were used in the experiments for each panel? What are 19/57 and 7/29? The text says "32.8% (19/58) and 40.5% (15/37)" (Lines 116 and 117). Fig. S1a needs the visible Y axis.

10. The weaning day in Fig.1 Moreover, "two inoculations (M), then Day 1 (Day 1)" (Line 104) must be " two inoculations (mid), then Day 1 (D1)". I could not find "PI" in Fig. 1 although the text says "post-inoculation (PI)" as a label.

11. The p values and the permutation number of PERMANOVA were missing in the text and/or Figures S2 and S3.

12. Figure 3c shows bars and SD. I believe that the Bray-Curtis index values are non-parametric as the authors used Mann-Whitney test. Please use a box and whisker plot in Figure 3c. Similarly, some data of Figure 2 seem to be non-parametric, and presentation with means and SD is inappropriate. Please use box and whisker plots in Figure 2.

6. PLOS authors have the option to publish the peer review history of their article (what does this mean?). If published, this will include your full peer review and any attached files.

Reviewer #1: No

Reviewer #2: No

---

## [Author Response · Author response to Decision Letter 0]

18 Jun 2020

To whom it may concern,

We sincerely appreciate the careful and thorough reading of our manuscript (PONE-D-20-08853) titled, “Interactions of Segmented Filamentous Bacteria (Candidatus Savagella) and bacterial drivers in colitis-associated colorectal cancer development” by the reviewers. They have made excellent suggestions and we have revised the manuscript accordingly. Below, we list each of the reviewers’ comments and suggestions individually, followed by our specific responses in bold text.

Reviewer #1: Wolfe et al. reported in this manuscript that CRC incidence, disease scores, tumor scores, and inflammation scores after Helicobacter spp. inoculation in Smad3-/- mice were significantly reduced in male mice compared to female mice only when SFB was inoculated. In addition, the authors showed that SFB inhibited colonization of Helicobacter spp. Relative abundance of Helicobacter spp. was predictive of CRC development in SFB- mice whereas E. Coli were significantly more abundant in mice which eventually developed CRC in SFB+ mice. Although the study was well planned and revealed interacting phenomena regarding intestinal bacterial colonization during CRC development in the Helicobacter spp-Smad3-/- mouse model, it was too descriptive. More rigorous examination needs to be performed. Specific comments are below.

1. Immune cell population and activation including expansion of Th17 cells and production of Th17 cytokines need to be tested in male and female mice in the presence/absence of SFB.

Thank you for this excellent suggestion. The purpose of this study was to examine the role of the gut microbiota longitudinally through the course of CRC development, to delineate bacterial taxa associated with differential susceptibility to CAC initiation, or differential disease severity (i.e., progression). Thus, CAC was our primary outcome measure for this study. While we intend to validate the expansion and function of several leukocyte subsets and relevant cytokines in future studies, it was beyond the scope of the current studies. While IL-17 is a logical downstream effector molecule induced by SFB, previous data from our lab suggest that CAC in Smad3 mice is driven by innate immune responses, while Helicobacter induces Th1 responses and mucosal barrier defects. Thus, our intent is to perform single cell RNAseq using GI tissue, MLN, and spleen of select groups of mice identified in this and a separate study. 

2. There were no differences in GM stability on the basis of sex in either SFB+ or SFB- mice (Fig. S4). Since the authors showed that CRC development was reduced in male SFB+ mice, this observation seems to contradict with the conclusion “Relative abundance of Helicobacter spp. was predictive of CRC development in SFB- mice”.

The stability of the GM is based on community-wide changes in beta-diversity, and increases or decreases in abundance of individual taxa, even dominant ones like Helicobacter, do not necessarily correlate with overall stability. Indeed, we respectfully maintain that the relative abundance of Helicobacter spp. is predictive of CRC development in SFB negative mice, but not SFB+ mice. There was no disease score difference in SFB- mice by sex, thus we do not believe this to be a contradiction. We apologize in advance if we have misunderstood the Reviewer’s comment and welcome further discussion if this is the case.

Minor comment

3. This reviewer suggests to present representative pictures of intestinal histology in addition to quantified results presented in the manuscript.

The reviewer makes an excellent suggestion and we have added a representative photo as Fig 1Bi and ii, and shifted the original Fig 1B to Fig 1C. Figure legends and text have been updated to reflect this.

Reviewer #2: Wolfe et al., described the protective role of SFB on colorectal cancer (CRC) of smad3-/- male mice. However, it is unclear that the interactions among three bacteria, SFB, Helicobacter and E. coli are relevant to the disease severity of CRC, because key and comprehensive analyses are missing in this manuscript. More detailed comments are provided below:

This study was originally performed to determine the effect of SFB, a common commensal in the mouse gut with potent immunomodulatory capacity, on the CAC phenotype, as its impact has been observed and studied in other disease models. The observations reported in the current study (i.e., interactions between Helicobacter spp., family Enterobacteriaceae, and SFB) were somewhat unexpected and, admittedly, somewhat observational. Nonetheless, we believe that our findings are very informative in terms of gut microbial ecology during disease initiation and progression as few (if any) studies have provided robust longitudinal data throughout all stages of CAC. Many of the ostensible interactions reported in the current data are of great interest to us, and will be tested prospectively in future studies to better determine causality and the mechanistic relevance to disease. 

While we would maintain that family Enterobacteriaceae and Helicobacter spp. behave differently within the context of SFB+ mice versus SFB- mice, we completely understand the Reviewer’s concern and have adjusted the verbiage in the manuscript to avoid over-interpretation. 

Comments:

1. It is unclear what Figure 3a shows. The text says " relative abundance of operational taxonomic units (OTUs)" (Line 169), but the figure has " Enterobacteriaceae" (family) and "Helicobacter spp." (genus), implying that the analysis is at family or genus level. Which taxonomic level did the authors use, OTU@>97%, genus, or family? If it is possible, please use yellow and blue to make the highlighted taxa clearer in the panel. Moreover, description of other OTUs is missing. I would like to request the supplemental table of OTU abundance of individual samples to review and judge the results of figures in this manuscript.

We appreciate the reviewer’s suggestion and apologize for the confusion. Family Enterobacteriaceae, now highlighted in light blue, comprised solely Escherichia-Shigella, Escherichia sp. The manuscript has been updated to specify this. Family Enterobactericeae is used as an umbrella term due to poor sequencing resolution between species within this family.

Helicobacter spp., now highlighted in orange-yellow, comprised Helicobacter uncultured bacterium and Helicobacter Ambiguous taxa. The vast majority (visible portion) of the orange-yellow bar of the barchart represents Helicobacter ambiguous taxa. The linecharts in Fig 4a and Fig S4a represent sums of both of the Helicobacters spp. present.

Arrows have been added to the figure to point out the light blue and orange-yellow portions within the bar charts.

We have also included the color-coded list of other OTUs present within the barchart as a new Fig S3, and similar Supplemental Figures have been relabeled.

2. For the conclusion "Three patterns emerged from pre-inoculation to 2W PI within the SFB-CRC-, SFB-CRC+, SFB+CRC-, and SFB+CRC+ groups. First, the GM of SFB+ mice, regardless of CRC development, remained relatively consistent from pre to 2W post inoculation when compared to the GM of SFB- mice. Second, colonization with Helicobacter spp. peaked at D4 in SFB- mice regardless of CRC development, but was blunted in SFB+ mice. Lastly, unresolved microbes in the family Enterobacteriaceae bloomed concurrently with Helicobacter spp. in all groups but SFB+CRC-." (Lines 170 - 176), the authors should perform multiple statistical analyses including LEfSe and FDR.

We appreciate the reviewer’s suggestion and have included additional LEfSE analyses for timepoints Pre and D4 of SFB+CRC+, SFB+CRC-, SFB-CRC+, and SFB-CRC- groups (4 classes per timepoint) as Fig S7 and added description of those figures within the text. The header “Family Enterobacteriaceae and SFB interact in CRC development” has been changed to “Family Enterobacteriaceae is predictive in SFB+ CRC development” to better reflect the observations within Fig 4B. 

3. The conclusion "Subjectively, SFB- mice appear to have greater community shifts following Helicobacter spp.-inoculation than SFB+ mice. " (Lines 188 - 190) and following statement require the result of statistical analysis.

We agree completely. We clarified within the manuscript that the graphs in Figure 3C are the Bray-Curtis measurements generated from the line lengths in the PCoA of Figure 3B, and supported those claims with statistical tests. 

4. "On average, SFB- Pre and D4 time-points were more dissimilar than Pre and D4 time-points of SFB+ mice (p = 0.023, Mann-Whitney rank sum test)," (Line 191 to 193). The authors should show medians but not averages as the data was non-parametric as they used Mann-Whitney rank sum test. The comparison also needs controls, Bray-Curtis indexes of internal Pre and D4, because the difference might simply reflect the internal variations of Pre or D4, and it must be multiple comparison test and another test other than Mann-Whitney is required. I have a similar concern on Figure S4.

We have clarified that each value given represents the Bray-Curtis Dissimilarity Index value between Pre and D4 of each individual mouse rather than the mean value at the Pre and D4 time-points (thus accounting for inter-individual variability), and that these are then categorized by SFB status and analyzed.

5. The analysis of the sections "Relative Abundance of Helicobacter spp. at D4 PI predictive of CRC development in SFB- but not SFB+ mice." and Family Enterobacteriaceae and SFB interact in CRC development is biased for particular bacteria. The authors should perform Spearman ranking test of all (from higher to lower) taxons with CRC levels. This is the most critical point for the science of this manuscript.

The reasons that Helicobacter spp. and Family Enterobactericeae were chosen were because of stark observations seen within the barchart in Fig 3A. It makes sense to look at Helicobacter spp. as the provocateur of inflammation within the disease model, and Family Enterobactericeae as a driver and passenger to inflammation. Testing for these has been done with a Yes/No basis for CRC, as Spearman rank correlations rarely yield an R value stronger than ±0.3 when relative abundance is compared to disease score. However, we have included a chart of OTU and Family level Spearman Rank Correlations for SFB+ and SFB- groups on Pre, D4, and D7 PI as Table 1 and included verbiage within the manuscript. That being said, we believe that these data should be interpreted cautiously, as multiple testing of this magnitude can lead to an exceedingly high false discovery rate, especially with many of the r values as low as they are. 

6. "Colonies were identified using matrix-assisted laser desorption/ionization time-of-flight (MALDI-ToF)

mass spectrometry. Seven of the eight samples were identified as Escherichia coli, as expected (Figure

5A)" Because Figure 5A too much simplified the result of MALDI-TOF, the authors should attach the supplemental tables of identified bacterial molecules and describe which database was used for identification of the molecules. Moreover, the authors should describe justification for the reason why they used mass spec as genome sequencing of isolates gives more accurate and detailed results with low cost.

More information regarding MALDI-ToF was added to the materials and methods section regarding databases used. The information provided from a given isolate allow for taxonomic identification from a database based on the full surface protein spectrum, while no information is provided regarding the identity of specific protein markers. 

MALDI-ToF was chosen as an initial identification modality due to the substantially lower costs associated with screening multiple isolates at our institution. For a pair of promising isolates, we did perform whole genome sequencing to search for specific genes which may contribute to disease, as stated in the manuscript.

7. According to the abundance of Enterobacteriaceae in Figures 3a and 4, the fact that the result of Figure 4a showed the dominance of E. coli, suggests cloning bias of bacterial isolates. This raises the concern about bias in the conclusion.

As E. coli is easily culturable, in contrast to the many OTUs which cannot yet be cultured, this is a valid concern. Part of the rationale for these cultures was to better identify the members of family Enterobacteriaceae (particularly those falling under the Escherichia-Shigella sp. taxa) which might be proliferating, as there are many differences in strains that cannot be detected by 16S alone. We do not assume that we cultured every single individual strain of family Enterobacteriaceae, but are definitely intrigued by what we did isolate and intend on following up on these physical properties as well as the full-genome sequencing data from these two isolates. This will give us a better idea of E. coli-specific virulence factors found within the GIT of our Smad3-/- model as a starting point for mechanistic studies in the future.

8. What does the finding on hemolytic features of one E. coli strain (Fig, 5b) mean? Is this associated with the conclusions of this manuscript?

The alpha-hemolytic properties of E. coli are most commonly associated with invasive strains of E.coli rather than facultative ones. These strains have been found to play a role in intraperitoneal infections, which will be interesting to study in a mechanistic sense within our model. We have added verbiage concerning the implications of this finding in the Discussion. 

Minor comments:

9. The figure legend is missing. How many mice were used in the experiments for each panel? What are 19/57 and 7/29? The text says "32.8% (19/58) and 40.5% (15/37)" (Lines 116 and 117). Fig. S1a needs the visible Y axis.

We apologize for the omission and have fixed this discrepancy in the text. The figures were correct, and the discrepant numbers reflect the data prior to discarding the SFB-exposed group (members of the SFB+ group which did not test positive for SFB at any timepoint via PCR as outlined in the materials/methods.) We have double-checked the stats/figures to make sure that these numbers reflect the final and correct ratios.

10. The weaning day in Fig.1 Moreover, "two inoculations (M), then Day 1 (Day 1)" (Line 104) must be " two inoculations (mid), then Day 1 (D1)". I could not find "PI" in Fig. 1 although the text says "post-inoculation (PI)" as a label.

We have fixed these mistakes as well and clarified that weaning is at Day 21 within the figure.

11. The p values and the permutation number of PERMANOVA were missing in the text and/or Figures S2 and S3.

We have added the p values in the figure legends of Fig S2 and pairwise charts in Fig S3. 

12. Figure 3c shows bars and SD. I believe that the Bray-Curtis index values are non-parametric as the authors used Mann-Whitney test. Please use a box and whisker plot in Figure 3c. Similarly, some data of Figure 2 seem to be non-parametric, and presentation with means and SD is inappropriate. Please use box and whisker plots in Figure 2.

We appreciate the astute suggestion and have amended the mentioned graphs to box and whisker charts as requested.

---

## [Decision Letter · Decision Letter 1]

10 Jul 2020

Interactions of Segmented Filamentous Bacteria (Candidatus Savagella) and bacterial drivers in colitis-associated colorectal cancer development.

PONE-D-20-08853R1

Dear Dr. Ericsson

We’re pleased to inform you that your manuscript has been judged scientifically suitable for publication and will be formally accepted for publication once it meets all outstanding technical requirements.

Kind regards,

Hiroyasu Nakano, M.D., Ph.D.

Academic Editor

PLOS ONE

Additional Editor Comments (optional):

Reviewers' comments:

Reviewer's Responses to Questions

**Comments to the Author**

1. If the authors have adequately addressed your comments raised in a previous round of review and you feel that this manuscript is now acceptable for publication, you may indicate that here to bypass the “Comments to the Author” section, enter your conflict of interest statement in the “Confidential to Editor” section, and submit your "Accept" recommendation.

Reviewer #1: (No Response)

Reviewer #2: All comments have been addressed

2. Is the manuscript technically sound, and do the data support the conclusions?

Reviewer #1: Yes

Reviewer #2: Yes

3. Has the statistical analysis been performed appropriately and rigorously? 

Reviewer #1: I Don't Know

Reviewer #2: Yes

4. Have the authors made all data underlying the findings in their manuscript fully available?

Reviewer #1: Yes

Reviewer #2: Yes

5. Is the manuscript presented in an intelligible fashion and written in standard English?

Reviewer #1: Yes

Reviewer #2: Yes

6. Review Comments to the Author

Reviewer #1: The authors addressed the reviewer’s concerns. Although the study is superficial, the results will contribute to the progress of the research field.

Reviewer #2: The authors have addressed all my comments well and the revised manuscript is suitable to publish in PLOS ONE.

7. PLOS authors have the option to publish the peer review history of their article (what does this mean?). If published, this will include your full peer review and any attached files.

Reviewer #1: No

Reviewer #2: No

---

## [Editor Report · Acceptance letter]

15 Jul 2020

PONE-D-20-08853R1 

Interactions of Segmented Filamentous Bacteria (Candidatus Savagella) and bacterial drivers in colitis-associated colorectal cancer development. 

Dear Dr. Ericsson:

I'm pleased to inform you that your manuscript has been deemed suitable for publication in PLOS ONE. Congratulations! Your manuscript is now with our production department. 

Kind regards, 

on behalf of

Professor Hiroyasu Nakano 

Academic Editor

PLOS ONE